# Various Therapeutic Methods for the Treatment of Medication-Related Osteonecrosis of the Jaw (MRONJ) and Their Limitations: A Narrative Review on New Molecular and Cellular Therapeutic Approaches

**DOI:** 10.3390/antiox10050680

**Published:** 2021-04-27

**Authors:** Sung-Woon On, Seoung-Won Cho, Soo-Hwan Byun, Byoung-Eun Yang

**Affiliations:** 1Division of Oral and Maxillofacial Surgery, Department of Dentistry, Hallym University Dongtan Sacred Heart Hospital, Hwaseong 18450, Korea; drummer0908@hanmail.net; 2Graduated School of Clinical Dentistry, Hallym University, Chuncheon 24252, Korea; kotneicho@gmail.com (S.-W.C.); purheit@daum.net (S.-H.B.); 3Institute of Clinical Dentistry, Hallym University, Chuncheon 24252, Korea; 4Division of Oral and Maxillofacial Surgery, Hallym University Sacred Heart Hospital, Anyang 14066, Korea

**Keywords:** medication-related osteonecrosis of the jaw (MRONJ), osteomyelitis, osteoporosis, oxidative stress injury, maxillofacial pathology, metabolic bone diseases, bisphosphonate, antiresorptive agents

## Abstract

Medication-related osteonecrosis of the jaw (MRONJ) is one of the most interesting diseases in the field of maxillofacial surgery. In addition to bisphosphonates, the use of antiresorptive and antiangiogenic agents is known to be the leading cause. However, the exact pathogenesis of MRONJ has not been established, and various hypotheses have been proposed, such as oxidative stress-related theory. As a result, a definitive treatment protocol for MRONJ has not been identified, while various therapeutic approaches are applied to manage patients with MRONJ. Although the surgical approach to treat osteomyelitis of the jaw has been proven to be most effective, there are limitations, such as recurrence and delayed healing. Many studies and clinical trials are being conducted to develop another effective therapeutic modality. The use of some materials, including platelet concentrates and bone morphogenetic proteins, showed a positive effect on MRONJ. Among them, teriparatide is currently the most promising material, and it has shown encouraging results when applied to patients with MRONJ. Furthermore, cell therapy using mesenchymal stem cells showed promising results, and it can be the new therapeutic approach for the treatment of MRONJ. This review presents various treatment methods for MRONJ and their limitations while investigating newly developed and researched molecular and cellular therapeutic approaches along with a literature review.

## 1. Introduction

Medication-related osteonecrosis of the jaw (MRONJ) is one of the refractory diseases in the maxillofacial area. Since it was first reported by Marx in 2003 [1], there has been no gold standard treatment for MRONJ. At first, it was noticed under the name of bisphosphonate-related osteonecrosis of the jaw (BRONJ). However, as more drugs were reported to induce osteonecrosis of the jaw, it has been renamed MRONJ in the position paper of the American Association of Oral and Maxillofacial Surgeons (AAOMS) [2]. Afterward, the definition of the disease and the classification of the stages were further refined. With the increase in the elderly population, the use of drugs that may cause MRONJ continues to increase. However, the development of treatment for MRONJ remains stagnant despite this trend. This may be attributed to the fact that neither the exact pathogenesis of MRONJ nor the definitive pathognomonic biomarkers have been identified. As a result, various studies and research for a precise diagnosis and treatment of MRONJ are continuously in progress.

Surgical treatment is considered to be the most standard treatment of MRONJ, although some differences exist in the treatment approach depending on the stage. Since the possibility of recurrence due to bone remodeling changes caused by the remaining MRONJ-inducing drugs in the body cannot be entirely excluded, therapies focusing on this point are being studied. Various treatment modalities such as hyperbaric oxygen (HBO) therapy, ozone therapy (OT), and low-level laser therapy (LLLT) have been suggested. However, they are not considered the primary therapeutic modality but are rather considered an adjunctive treatment. At present, studies on molecular and cellular approaches targeted to hypotheses that are presumed to be the mechanism of MRONJ are being conducted. Among them, teriparatide has recently been proven effective through a randomized controlled trial [3], showing the possibility as a promising therapeutic agent in the future. Furthermore, bone morphogenetic protein (BMP) and autologous platelet concentrates (APC) are applied in clinical and research environments as adjunctive agents. Mesenchymal stem cell (MSC) therapy is also drawing attention as another therapeutic approach.

This review aims to present various therapies for MRONJ and its limitations. In addition, it examines newly developed and researched molecular and cellular therapeutic approaches along with a literature review.

## 2. Staging and Possible Hypothesis of the Pathogenesis of MRONJ

### 2.1. Staging of MRONJ and Treatment Strategies

The AAOMS first presented the staging system for BRONJ to establish a reasonable treatment guideline and collect data to evaluate patients’ prognosis with the history of bisphosphonate usage in 2007 [4]. At this point, BRONJ was classified into at risk, Stage 1, 2, and 3, with several clinical traits presented for each stage. However, as the demand for further classification had been raised, the first staging system had to be changed. As a result, Stage 0 was added to the staging system in 2009 [5]. Although the risk of progressing from Stage 0 to a higher stage was still unknown at that time, it was necessary to include patients presenting with nonspecific symptoms or clinical findings that may be due to bisphosphate exposure. Besides, the definition of Stage 3 was revised in this updated position paper to include advanced maxillary diseases such as oroantral/oronasal communication, exposed and necrotic bone extending to the maxillary sinus, and osteolysis extending to the sinus floor [5]. As knowledge and experience of BRONJ accumulated, and as other drugs thought to induce osteonecrosis were added, a new position paper by AAOMS was proposed in 2014 [2]. From this time on, the name MRONJ began to be commonly used since other antiresorptive and antiangiogenic agents besides bisphosphonate were included as drugs that could induce osteonecrosis. The drugs known to be associated with MRONJ are shown in Table 1. Although there was no remarkable change in staging, cutaneous or mucosal fistula, which is probed to the bone, was included in the definition of the exposed bone and was added as a clinical manifestation in Stages 1, 2, and 3. Additionally, the revised position paper stated that Stage 0 in the previous position paper was appropriate in that it was possible to detect patients with the prodromal disease. Therefore, Stage 0 was maintained in the staging system. The latest staging system suggested by AAOMS is presented in Table 2, and the clinical cases for Stages 0, 1, 2, and 3 are shown in Figure 1, Figure 2, Figure 3 and Figure 4, respectively.

### 2.2. Possible Hypothesis of Pathogenesis

The pathophysiological mechanisms of MRONJ have not been fully elucidated, although the first MRONJ cases were reported nearly 20 years ago. It is estimated that inhibition of osteoclastic activity in bone resorption and remodeling and/or inhibition of angiogenesis, which are related to the function of the osteonecrosis-induced drugs, plays a significant role in the occurrence of MRONJ. However, there are various hypotheses concerning the pathogenic mechanism of MRONJ.

#### 2.2.1. Suppression of Bone Remodeling

Antiresorptive agents, including bisphosphonate and denosumab, inhibit osteoclast differentiation and function, increase apoptosis of osteoclast, and finally lead to decreased bone remodeling and resorption [6,7,8,9,10]. These conditions may be affected by triggering factors such as dental surgery, including tooth extraction, microtrauma to the jaw, and local inflammation, resulting in necrosis and jaw bone exposure [11]. It is not clear why MRONJ frequently occurs in the maxilla and mandible. The high rate of bone remodeling in the jaw may predispose to osteonecrosis compared to other bones. Since the bone turnover rate of the alveolar bone is more than ten times faster than that of the long bone, it seems to be related to the ability to contain much more bisphosphonates in the alveolar bone compared to the bones in other sites [12]. Although there are objections that bone turnover is not reduced in osteonecrosis lesions [13], and active bone resorption occurs due to the presence of osteoclasts in the osteonecrotic areas [14,15], this hypothesis is further strengthened by the favorable results of teriparatide, which stimulates the osteoclast activity [3,16].

#### 2.2.2. Inhibition of Angiogenesis

Inhibition of angiogenesis is another central hypothesis that is thought to be a possible cause in the pathogenesis of MRONJ [17,18,19,20]. Because the interruption of the blood supply has classically been thought to cause osteonecrosis, it is natural for this hypothesis to draw attention as the leading cause of MRONJ. However, the exact etiopathogenesis of MRONJ associated with antiangiogenic agents is not well known. It is presumed that angiogenesis inhibition adversely affects the bone regeneration capacity after bone damage, delays bone remodeling or healing, and may increase susceptibility to superinfection [21,22]. Besides, antiangiogenic agents have other effects that may be presumed to be the cause of MRONJ. The anti-vascular endothelial growth factor (VEGF) agents such as bevacizumab and rituximab can inhibit osteoblast differentiation and macrophage chemotaxis [23], and the tyrosine kinase inhibitors (TKI) such as sunitinib and sorafenib act on macrophages and interferes with the osteoclast differentiation [24]. Interestingly, zoledronate can also cause a decrease in the circulating VEGF levels, leading to a reduction in angiogenesis [25]. However, it is thought that denosumab, another antiresorptive agent, does not cause inhibition of angiogenesis.

#### 2.2.3. Infection and Inflammation

There is an opinion that the jaw is relatively vulnerable to infection because it is separated from oral pathogens with the help of only a thin mucoperiosteum, unlike other bones protected by thick soft tissues and skin [26]. Indeed, several animal studies have shown that the combination of inflammation or bacterial infections with systemic antiresorptive agents causes osteonecrosis [27,28,29,30,31,32]. However, in the progression of MRONJ, it is unclear whether osteonecrosis occurs first or whether infection and the resulting inflammation first occur. Although bisphosphonates have been shown to have bacterial biofilm-forming effects in osteonecrosis lesions [13,33], no specific infection focus was observed in most patients with MRONJ. Therefore, it is difficult to clearly state whether infection and inflammation are essential factors for osteonecrosis occurrence and progression.

#### 2.2.4. Toxicity to Soft Tissues

Bisphosphonates, in addition to their effects on the osteoclast, have direct toxicity to soft tissues. Proliferation and transportation of oral keratinocytes can be inhibited by bisphosphonates [19,20,34], resulting in persistent bone exposure and infection. Therefore, trauma or procedures that cause intraoral wounds may be associated with this toxicity and induce osteonecrosis. However, since most bisphosphonates are excreted through the kidneys several hours after reaching the bloodstream, the concentration of bisphosphonates in soft tissues has been reported to be lower than that in bone [35]. Therefore, this hypothesis needs to be further elucidated.

#### 2.2.5. Oxidative Stress

Another hypothesis for the pathophysiological mechanism of MRONJ is an oxidative stress-related theory. Oxidative stress refers to tissue damage caused by the loss of balance between reactive oxygen species (ROS) within tissue/cell and antioxidant mechanisms, resulting in a relatively excessive production of ROS. ROS target proteins, unsaturated phospholipids, and nucleic acids of DNA or RNA can weaken a cell’s structure and viability [36]. Therefore, it is not surprising that oxidative stress has been reported in cancer patients with advanced metastasis [37]. Moreover, it is well known that bisphosphonate treatment can cause oxidative stress [38] and that persistent local inflammation and associated infection processes can produce ROS [39,40,41].

Oxidative stress is related to osteonecrosis, mainly the one induced by steroids [42,43,44]. Since steroids in isolation were not regarded as a factor inducing MRONJ, oxidative stress was not initially presumed to be the inducing mechanism of MRONJ. However, the possibility of an association between oxidative stress and MRONJ has been suggested due to the relationship between bisphosphonates and oxidative stress. Starting with a study by Bagan et al. [45] in 2014, which reported that oxidative stress was detected in the MRONJ patients, animal experiments and clinical studies focusing on oxidative stress as a pathophysiological mechanism of MRONJ have been actively conducted in recent years. It is noteworthy that oxidative stress and bisphosphonate may induce osteonecrosis of the jaw after an invasive dentoalveolar surgery in mice [46] and that bisphosphonate-induced ROS could delay wound healing by preventing the growth and migration of oral fibroblasts [47]. Besides, it is interesting that hypotaurine, which acts as an antioxidant associated with cellular defense against oxidative stress, was significantly elevated in patients’ saliva with MRONJ and thus showed potential as a new biomarker for the early detection of MRONJ [48]. It is also important to note that [4-(methylthio) phenylthio] methan bisphosphonate, the most potent antioxidant among bisphosphonates, exhibited fewer adverse effects in patients with inflammatory bone disease, including rheumatoid arthritis and periodontitis, showing the potential as a “rescue” bisphosphonate [49]. At present, it is the beginning step of finding out that oxidative stress may cause MRONJ. However, the accurate identification of the relationship would be worthwhile because the antioxidants that can reverse oxidative stress can be applied to treat the MRONJ.

## 3. Various Therapeutic Methods for MRONJ

### 3.1. Medication

Medication, which may be referred to as conservative medical treatment, is the primary method currently available in the early stages of MRONJ. According to the position paper of AAOMS in 2014 [2], patients with MRONJ Stage 0 and 1 can benefit from medical treatments such as systematic antibiotics and/or antimicrobial rinse. It can also be applied as adjuvant therapy in Stages 2 and 3 when evidence of infection begins to appear. However, the conservative medical treatment may take the lead even in later stages that require surgical treatment despite these recommendations. For example, if the patient is reluctant to undergo surgery or the patient’s general condition may not allow surgery, conservative medical treatment might be a good alternative [50].

The most widely used antibiotics for the systematic medication in patients with MRONJ appear to be penicillin, amoxicillin, amoxicillin/clavulanate, metronidazole, or a combination thereof [51]. These drugs are also used as medications for common oral infections. Although the association between oral microflora and MRONJ has not been established, it is thought that these drugs are being applied because bacterial migration in the progression of MRONJ can cause inflammation or infection [52]. When it comes to local antimicrobial use for the management of MRONJ, there seems to be no disagreement that chlorhexidine is the first choice.

Stage-specific therapy has been a consensus among clinicians, and surgical treatment is considered effective treatment at the later stages of MRONJ. However, there is an opinion that conservative medical treatment can be applied at all stages. Ramaglia et al. [53] recommended starting with two weeks of conservative medical treatment for the management of MRONJ in their systematic review and meta-analysis because it showed effectiveness at all stages. Although the medical treatment had a lower treatment success rate in the advanced stages, 13 selected studies in Ragmalia et al.’s review [53] showed success rates of 63.6–100% in Stage 2 and 73% in Stage 3. Another advantage of conservative medical therapy is that it may also reduce the possibility of unnecessary surgical treatment by preventing progression to the higher stage of the disease. Besides, it may relieve the wound, even if it cannot achieve a complete cure of MRONJ, making it possible to perform less invasive surgery when performing a surgical treatment.

Medication therapy still has some limitations. First, there is no consensus on which antibiotic is the most effective and the total duration of application. Besides, since most of the papers did not specifically indicate the dosage of antibiotics, the dosage for each antibiotic has not been established. Second, although conservative medical treatment often works in the advanced stages, most clinicians agree that it only serves an auxiliary role to control infection in the later stages. They claim that, eventually, surgical treatment is necessary for the complete healing of the MRONJ wound. Lastly, there is a possibility of adverse events that may arise from the long-term use of antibiotics. Possible adverse events from long-term use of antibiotics consist of gastrointestinal events, end-organ toxic effects, multidrug-resistant organism colonization, and Clostridium difficile infections [54,55]. Although the probability of these adverse drug events is low, clinicians should consider this risk and apply long-term use of antibiotics with care.

Although the efficacy and scope of application have not been fully established, conservative medical treatment is a safe method that can be tried at first and remains the most commonly applied method in the treatment of MRONJ.

### 3.2. Surgical Treatment

Surgical treatment is known to be an essential method for the management of the later stages of MRONJ. According to AAOMS in 2014 [2], surgical treatment such as debridement was suggested from Stage 2, and its necessity was emphasized. The surgical treatment for MRONJ consists of debridement, curettage, sequestrectomy, and resection with or without microvascular reconstruction [2,56]. Although there are some differences according to the literature, debridement of the soft tissue, curettage of the bone with no extension and sequestrectomy belong to conservative surgical treatment, and major or complete resection of the necrotic lesion with or without microvascular reconstruction belongs to aggressive surgical treatment [57,58].

As mentioned above, surgical treatment is close to the most standard treatment in the management of MRONJ. Its effectiveness has already been widely proven in Stages 2 and 3 [59,60,61,62,63]. Recently, a prospective cohort study has reported that it is also useful in early stages such as Stage 1. Giudice et al. [64] evaluated the time to restore mucosal integrity and downstage lesions after surgical treatment in 57 patients with Stage 1 and 72 patients with Stage 2. Compared to patients with Stage 2, the time taken for mucosal coverage in patients with Stage 1 was significantly shorter (56.4 ± 54.5 days for MRONJ Stage 1 vs. 83.1 ± 74.9 days for MRONJ Stage 2) [64]. However, the time taken to achieve downstaging was shorter in Stage 2 than in Stage 1 patients, and this is presumed to be because Stage 2 patients can achieve downstaging even if the exposed bone is not completely removed and only the infection disappears, but Stage 1 patients must show complete wound healing for the downstaging.

In summary, Giudice et al. [64] suggested that surgical therapy has a high success rate and a short treatment period in the early stages of MRONJ. Surgical treatment provides pain and infection control, relief of soft tissue irritation, and reduced osteolysis [56,65]. In addition to the therapeutic effect of surgical treatment, surgical treatment has an additional advantage that enables histopathological examination of the removed tissues to exclude metastasis in oncologic patients in whom primary neoplasm is possible [66].

In the treatment of MRONJ, the extent of surgical treatment and which surgical method is most useful have not yet been established. Recently, a systematic review was published comparing conservative and aggressive surgery [58]. As a result of reviewing 13 studies, extensive resection to have a viable bleeding margin showed the best results in patients in Stage 3 [58]. When comparing the aggressive surgery and simultaneous microvascular flap reconstruction with the aggressive surgery only, the former showed complete mucosal healing (97% vs. 85%) [58]. However, since such surgery may cause more severe morbidity in patients, a judgment considering risk/benefit is required. There is no study comparing the effectiveness of conservative surgery and aggressive surgery in Stage 2.

Unique techniques worth mentioning in surgical treatment are the buccal fat pad flap in the maxilla and the mylohyoid flap in the mandible. In the surgical treatment for MRONJ, removal of the necrotic bone and soft tissue covering using flaps are required. In that case, if the closure is performed using only local mucosal flaps, a high failure rate is shown, which is due to poor vascularity of the MRONJ lesion [67]. Therefore, microvascular flaps can be used when a large soft-tissue defect is expected, but it cannot always be the first option due to donor site morbidity and the prolonged operation time. As a suboptimal solution, buccal fat pat flap or mylohyoid flap could be used, and when an appropriate case selection was accompanied, favorable results with minimal complications and no recurrence were shown [68,69].

Surgical treatment with additional surgical tools has been reported. Ultrasonic bone surgery, also known as piezoelectric surgery, has been actively performed in the field of maxillofacial surgery, and it has the advantage of enabling less invasive bone cutting without damaging soft tissues. Blus et al. [70] reported a complete soft tissue closure within postoperative one month and no recurrence during a 10–54 month follow-up period after performing surgical removal of necrotic bone and debridement using a piezoelectric device in 18 patients with MRONJ. In addition to piezoelectric surgery, there is also autofluorescence-guide bone surgery in which surgery is performed using fluorescence detection devices to determine the resection margin of necrotic bone during surgery. Since the incomplete sequestrectomy can cause recurrence, complete removal of the necrotic bone is essential during surgical treatment for MRONJ, and it is not easy to determine the margin of osteonecrosis. To overcome these difficulties, Giudice et al. [71] published a prospective study comparing the efficacy of surgery using the VELscope (visually enhanced lesion scope) fluorescence lamp and conventional surgery. However, autofluorescence-guided surgery did not show superior results to conventional surgery in mucosal healing and the quality of life [71]. As another method, a case report was recently published that applied dynamic navigation for the surgical treatment of MRONJ [72]. Dynamic navigation is a technology that enables the tracking of the position of surgical instruments while viewing the cone-beam computed tomography images in real-time and has also recently been applied in the field of dental implants. Pellegrino et al. [72] reported a case of successfully treating a patient with MRONJ Stage 2 involving inferior alveolar nerves by combining dynamic navigation with ultrasonic surgery and suggested that more clinical trials are needed to evaluate the effectiveness and safety of this technology in the management of MRONJ. Although surgical treatment using these additional surgical tools mentioned above requires more studies and evidence, it is meaningful that the surgeon can perform a more straightforward and comfortable operation in the surgical treatment for MRONJ by using these tools.

There are some disadvantages to the surgical treatment, although it has been proven to be effective. Most of those are related to the operation itself and are associated with complications after surgery. Some authors reported aggravation of the symptoms, pathological fractures, and loss of parts of the jaw after the surgical treatment [73,74]. However, these complications are preventable, and there is no need to fear or hesitate to perform surgical treatment when the surgery is indicated.

Conclusively, surgical treatment has not yet been proven to treat patients in the early stages of MRONJ, but it has been established as a necessary treatment for later stages. Attempts on surgical equipment and surgical methods for accurate and safe surgery are in progress, and attention is paid to continuing follow-up studies.

### 3.3. Application of Regenerative Materials

In general, the application of specific materials is additionally performed during the surgical treatment of MRONJ. However, it is necessary to discuss the types, mechanisms, and effects of these materials separately. Currently, regenerative materials widely used for the treatment of MRONJ include BMP and APC.

BMP-2, a group of signaling molecules that belongs to the superfamily of transforming growth factor-β (TGF-β), has been used in orthopedic and maxillofacial surgery to increase bone remodeling. Due to its osteoinductive property [75], it is thought to have a potential effect of reversing the remodeling suppression in MRONJ and is currently being used to treat MRONJ [76,77,78]. BMP-2 is not generally applied as a single treatment modality but is applied in conjunction with surgical treatment, and it is delivered with a carrier to the defect from which the necrotic bone has been removed. Absorbable collagen sponges are frequently used as carriers for BMP-2, and some studies have shown that a rapid osteoinductive process is derived when these are applied to the jaw bone [79,80]. However, BMP-2 could result in some significant side effects such as swelling, inflammation, seroma, and carcinogenicity, although these could be dose dependent [81]. Therefore, it is considered that additional studies are needed to prove the efficacy and safety of BMP-2 further.

APC is also applied along with the surgical treatment of MRONJ, and its effect is achieved through the local application. APC is rich in several growth factors secreted from platelets, such as platelet-derived growth factor, TGF-β, endothelial growth factor, and vascular endothelial growth factors [82]. Therefore, since they can stimulate and accelerate tissue healing, the application of APC has been reported in the treatment of MRONJ. According to the definition of Dohan Ehrenfest et al. [83], APC is composed of 4 groups depending on the fibrin and leukocyte content. Among them, platelet-rich plasma (PRP) is the most frequently applied for treatment for MRONJ [84], and the application of leukocyte-rich and platelet-rich fibrin (L-PRF) has also been reported [85]. In particular, L-PRF has been in the spotlight as an adjunctive agent in recent years because it releases growth factors for a longer time at the application site compared to PRP. PRP is known to release 95% or more of the presynthesized growth factors within 1 h [86], while PRF is known to sustain the release of these growth factors for at least one week [87]. Besides, L-PRF has the effect of preventing infection by leukocytes and has the function of regulating the immune system [83]. According to a systematic review published in 2016 by Lopez-Jornet et al. [84], 85.98% of the complete resolution was reported when APC was applied with a combination of surgery to treat MRONJ. However, according to another systematic review by Fortunato et al. [88], the application of APC combined with surgery showed a better healing rate than the surgery alone (87.8% vs. 63.8%), but it was not statistically significant. Although sufficient scientific evidence to prove the effectiveness of APC in the treatment of MRONJ is lacking, it is considered promising as an adjunctive agent considering their local immunomodulatory properties and potential of accelerated tissue healing. The main limitation of APC is that there are no specific guidelines or protocols in the application. Besides, in the treatment of MRONJ, which is thought to be a bone disease, the question remains about the direct therapeutic effect of APC. These points will have to be solved in order for APC to be widely used for MRONJ.

### 3.4. HBO and OT

HBO and OT have been considered effective adjunctive treatment modalities in situations where normal bone healing is impaired, such as osteoradionecrosis and chronic osteomyelitis of the jaw [89,90,91]. Since MRONJ also exhibits the characteristics of reduced bone remodeling and impaired wound healing, HBO and OT are being applied with an anticipation that they might be helpful in the treatment of MRONJ.

HBO was traditionally thought to produce beneficial oxygen gradients [90,92] or correct pathogen killing of hypoxic-impaired leukocytes [89,93]. However, it was later discovered that HBO increases ROS and reactive nitrogen species (RNS), affecting the signaling processes important in wound healing [94,95,96,97,98,99,100,101]. ROS and RNS are known to influence osteoclast differentiation, activity and participate in regulating bone metabolism [102,103,104]. Along with this expectation, Freiberger et al. [105] published a case series in which HBO was applied to MRONJ patients, and remission or improvement was found in 62.5% of patients. Later, Freiberger et al. [106] performed a randomized controlled trial to evaluate HBO’s effectiveness based on their previous experience, but the HBO group did not show any significant difference from the control group in terms of complete gingival coverage. According to a recent systematic review [107], HBO mainly was applied as a neoadjuvant and/or adjuvant therapy in addition to the surgery rather than being used alone. Moreover, 5.17% of cases showed complete remission, while in the majority (48.27%) of the cases, slight benefits such as improvement or stability of symptoms were reported [107]. Therefore, it can be seen that HBO remains in its position as an adjunctive treatment method in the treatment of MRONJ at present.

Ozone generally exists in the form of gas composed of three oxygen atoms and positively affects hard and soft tissues by stimulating endogenous antioxidants [108,109,110]. Therefore, OT has been applied to various indications, including infectious disease, immune dysfunctions, ischemic pathologies, and neurodegenerative pathologies [108,109,110]. When ozone was applied intraorally, infection and inflammation were reduced in animal studies [108,111]. Agrillo et al. [112] first introduced OT as a new therapeutic protocol for patients with MRONJ by combining local minor curettage and antibiotic use. They noted that ozone exerts a positive effect on bone defects through oxidative preconditioning, which stimulates endogenous antioxidant systems and blocks the xanthine/xanthine oxidase pathway for ROS generation [112]. They also stated that ozone has beneficial effects on blood circulation by increasing the concentration of red blood cells and the rate of hemoglobin, which has a biological effect on oxygen, calcium, phosphorus, and iron metabolism germicidal properties [112]. After that, Agrillo et al. [113] reported that 131 patients with MRONJ had a 90% success rate after applying OT with minimally invasive surgery and antibiotic use. In a recent systematic review by Sacco et al. [107], OT was also most commonly used as neoadjuvant and/or adjuvant therapy related to surgery, as was HBO, and a complete resolution was reported in 44.58% of cases, and some symptom improvement was reported in 22.94%. Therefore, OT is also considered an adjunctive therapy for MRONJ treatment that requires more evidence at present.

The main limitation of HBO and OT is that there is no precise protocol besides the relatively low complete resolution rate. For HBO, 2.0–2.5 atm and 90–120 min have been reported for application pressure and application time [106,114,115], but it is currently unclear whether these values are appropriate and optimal. Moreover, guidelines for preoperative and postoperative application and the appropriate number of sessions have not been established. Likewise, in the case of the OT, the delivery method, dose, and frequency of ozone application were not established, and in many studies used, it was often unclearly mentioned. Agrillo et al. [112] initially applied ozone for 20 days at a frequency of application of twice a week for 5 min. In their subsequent study [116], one cycle of OT was performed each before and after surgery, and one cycle consisted of 8 sessions applied for 3 min. Ripamonti et al. [117] applied ozone oil to the lesion once every three days for 10 min and performed a maximum of 10 applications. The inconsistent results due to different application methods for each study can be a disadvantage of OT. Furthermore, in the case of HBO, special facilities such as a pressure chamber are required, and in the case of the OT, special equipment that delivers ozone gas is required. These are considered substantial obstacles to the wide use of HBO and OT for the management of MRONJ.3.5. Laser Therapy

Laser therapy has recently been used in the treatment of MRONJ in isolation or in combination with other treatment modalities, and its beneficial effects on tissue healing are the main reasons for its use [118,119]. The most widely used type of laser is LLLT, which is thought to be a promising adjuvant therapy in the treatment of MRONJ. It is known to regulate cellular metabolism, relieve pain, and improve wound healing. In addition, laser irradiation induces biostimulant effects that increase the number of differentiated osteoblasts and their activity, and the effect of LLLT on bone regeneration has also been reported in the field of dentistry [120]. Laser’s unique non-invasive properties and antibacterial effects on soft and hard tissues are also considered advantages of laser therapy.

As another kind of laser, the Er:YAG laser, which belongs to the family of high-power lasers, is thought to be a promising treatment modality for MRONJ. Because it has an affinity for water and hydroxyapatite and is well absorbed by them, it may be helpful in the management of bone tissues. More conservative surgery is possible because it produces an ablative surface conducive to cell attachment while less damaging the surrounding tissues [121]. When Er:Yag laser surgery was applied to the MRONJ lesion combined with LLLT, excellent mucosal healing was shown, and in the long-term follow-up, most cases showed complete healing [121,122]. Vescovi et al. [122] reported that the early laser-assisted conservative surgical approach with LLLT is a more reliable treatment than conservative medical therapy. However, since there were very few patients in Stage 3 treated with an Er:YAG laser in Vescovi et al.’s study [122], the application in the later stage of MRONJ seems to be lacking evidence.

Regarding the effectiveness of laser therapy, Weber et al. [123] published a systematic review in 2016 to investigate the positive effects of laser therapy in the management of MRONJ, and treatment modalities including laser therapy showed superior results compared to the conventional surgical therapy and/or the conservative medical therapy in terms of cure and improvement of lesions in the early stages of MRONJ. However, since they could not find a randomized controlled trial related to laser therapy, careful interpretation is considered necessary.

Among the limitations of laser therapy, the most critical is the lack of consistent reports on laser parameters. Due to the diversity of variables such as laser type, output power, output frequency, application time, and distance between the laser light source and the applied tissue, it is not easy to establish a standard protocol and collect results from every study. This may cause clinicians to hesitate to apply laser therapy to MRONJ treatment. Moreover, the need for a separate laser device and the need to wear safety glasses to protect the eyes from lasers are annoying disadvantages of laser treatment. If these shortcomings are addressed and additional results of higher-level research accumulate, laser therapy may play a more significant role in the treatment of MRONJ.3.6. Drug Holiday

As much as the debate over effective MRONJ treatment modality, the drug holiday is provoking a fierce debate among clinicians. Strictly speaking, drug holidays cannot be a treatment modality, but it is worth mentioning in this paper because it is an essential consideration in the prevention or treatment of MRONJ. A position paper of AAOMS in 2009 recommended discontinuation of oral bisphosphonates for three months before and three months after invasive dental surgery [5]. However, in a later position paper in 2014 [2], the modified drug holiday strategy was suggested by Damm and Jones [124]. That is, for patients who have been administered bisphosphonates for an extended period of more than four years, it is suggested that having a drug holiday for two months before invasive dental procedures is theoretically advantageous. Besides, drug holidays of the same period were recommended for patients who were administered bisphosphonate for less than four years and were also administered corticosteroids or antiangiogenic agents. Since it has been reported that a high total accumulation dose of bisphosphonate is associated with a significantly poor prognosis in the treatment of MRONJ [125], a drug holiday is considered to be necessary as the duration of bisphosphonate administration increases. However, there is no data on a drug holiday for denosumab and antiangiogenic agents, so, in this case, the clinician faces a difficult situation. Moreover, there is no explicit mention of the postoperative drug holiday in the 2014 position paper [2]. It only states that antiresorptive therapy should not be resumed until osseous healing has occurred. In the surgical treatment of MRONJ, the postoperative drug-free period is also crucial in that postoperative antiresorptive therapy or antiangiogenic therapy must be temporarily stopped not only because the bone exposure is accompanied during surgery but also because the bone healing must be expected after surgery.

There are two main views on the drug holiday for MRONJ: effective [126,127,128] or ineffective [129,130,131]. A recent systematic review [132], including three prospective studies and 11 retrospective studies, concluded that a drug holiday discontinuing high-dose antiresorptive therapy was unnecessary. However, since the studies included in the previous systematic review were heterogeneous, the meta-analysis could not be performed, and there is a limitation that the conclusion of only one controlled prospective study was asserted as it is. Therefore, at present, it is not possible to conclude whether or not a drug holiday is necessary for the management of MRONJ and the exact duration of suspension for each drug. Additional position papers based on data accumulation from more studies are expected in the future.

## 4. Promising Molecular and Cellular Therapeutic Methods for MRONJ

### 4.1. Teriparatide

Teriparatide is an analog of the human parathyroid hormone (PTH), consisting of the first 34 amino acids produced by genetic recombination. It was first approved for use in the treatment of osteoporosis by the United States Food and Drug Administration in 2002. When administered intermittently at low doses, PTH has an osteoanabolic effect, but when applied continuously, it exhibits an osteocatabolic effect, as seen in hyperparathyroidism [133]. In addition to its basic anti-fracture effect, it is widely used because its efficacy has been proven in low bone turnover conditions seen in patients with diabetes or patients undergoing long-term use of bisphosphonates [134,135,136].

In MRONJ, the expected effects of teriparatide might be to activate bone remodeling and increase bone formation, thereby inducing bone healing and necrotic bone removal. There is also an opinion that teriparatide shows its efficacy by activating WNT signaling and suppressing sclerostin production [133]. The osteoanabolic effect of teriparatide was confirmed in the craniofacial area [3,16], and in the field of dentistry, it was also found to promote bone growth and healing in patients with chronic periodontitis [137]. Due to this tangible osteoanabolic effect, expectations for the application of teriparatide for the management of MRONJ are increasing.

Among the studies that applied teriparatide for the treatment of MRONJ, the most interesting study is the randomized controlled study conducted by Sim et al. [3] in 2020. In their study, 34 patients with a total of 47 MRONJ lesions randomly received teriparatide (20μg/day) injection or placebo injection for eight weeks and were observed for 12 months. Teriparatide healed 45.4% of lesions at 52 weeks, showing a higher resolution rate than the placebo group in which 33.3% of lesions were healed. Teriparatide also increased bone volume, reducing bone defect size in a larger proportion of patients at 52 weeks (80.0% teriparatide versus 31.3% placebo). Besides, teriparatide increased the levels of P1NP and CTX, which are bone turnover markers. P1NP increased by three times compared to placebo at 4 and 8 weeks after receiving teriparatide, and CTX increased by 30% at eight weeks after. Sim et al.’s study [3] is significant because it is the first study to apply teriparatide in the treatment of MRONJ with a double-blind, randomized design. It also presents a new perspective and instills hope in the treatment of MRONJ.

Even with this promising teriparatide, there are still several aspects to overcome. In a toxicity study using rats [138], teriparatide increases osteosarcoma incidence, and there is a concern about the potential risk for osteosarcoma. However, it is difficult for this result to be applied equally to humans since much larger doses have been administered throughout their entire life (about 24 months) than those applied to humans. Besides, these findings were not confirmed in the subsequent studies of rats and monkeys with long-term treatment [139,140]. Moreover, during the period of use after being marketed for nearly 18 years, according to data from clinical studies or post-marketing surveillance, the association between teriparatide treatment and osteosarcoma has not been observed in humans [133,141].

Nevertheless, since the osteoanabolic effect of teriparatide can theoretically cause the proliferation of dormant malignant cells in bone, it is considered that the safety of its use in oncologic patients should continue to be verified. Second, there are no results for long-term treatment. Therefore, large-scale and high-quality clinical studies about the long-term effect of teriparatide are continuously needed. Furthermore, the relatively high price of teriparatide and the need for good patient cooperation due to subcutaneous injection may be a limitation in the widespread use of teriparatide. Therefore, the use of teriparatide for the treatment of MRONJ should be determined based on an individual patient risk assessment by clinicians and should be applied in consideration of proven beneficial effects and potential safety issues.

### 4.2. MSC Therapy

MSCs are multipotent cells that have been widely applied in regenerative medicine in recent decades. MSCs are involved in organogenesis during embryogenesis, and after that, in the maintenance of adult tissues.

It is well known that MSCs can differentiate into tissue-forming cells, such as osteoblasts, chondrocytes, and adipocytes. Based on their capacity to differentiate into osteoblasts and their immunomodulatory properties, MSCs might be used as a graft material for osteonecrosis areas [142]. MSCs are generally obtained from bone marrow or adipose tissue in adults, but they also exist in various other tissues [143].

MSCs have been mainly applied in the form of grafts to verify the therapeutic potential of MRONJ, and MSC grafts have obtained good results in mice, pigs, and humans [144,145,146]. However, these results seem to be due to their immunomodulatory properties rather than their ability to differentiate into osteoblasts. The efficacy of an MSC graft was associated with its ability to increase IL-10, TGF-β1, and regulatory T cells and decrease IL-6, IL-17, and c-reactive proteins [145,146]. There is no evidence that MSC grafts are directly related to bone regeneration by their osteoblast differentiation. Since the average survival time of the grafted MSCs is from 1 to 2 weeks, this period is insufficient for direct bone regeneration, but rather it is estimated to be indirectly related to bone regeneration [147]. The indirect effect of MSC graft is presumed to stimulate regional endogenous cells to differentiate into osteoblasts, thereby affecting bone regeneration [2]. It was found that the jaw bone cells arise from cranial neural crest stem cells (NCSCs), and the jaw defects are recovered by neural crest cell recruitment [148]. Therefore, it is thought that graft-containing NCSC is beneficial for the healing of jaw lesions, and it was recently shown that adult bone marrow MSC and adipose-derived MSC are populations containing NSCS. The effectiveness of MSCs related to NSCS is a part that requires further research, but it is worth continuing studies.

Although there are many animal studies to investigate the effect of MSC on MRONJ treatment, few studies have applied MSC to humans to treat MRONJ, and only studies at the level of the case report or pilot study have been reported. Cella et al. [144] first published a case report in 2011 that achieved complete healing of the lesion after applying autologous bone marrow stem cells to a patient with MRONJ Stage 3 who did not respond to the conventional treatment. They injected bone marrow stem cells harvested from the posterior superior iliac crest in a fibrin sponge carrier, along with PRP, and applied it to the bone lesion. The resolution of symptoms and improvement of the lesion were obtained after two weeks, while bone healing and concentric ossification were observed through computed tomography at postoperative 15 months. Complete healing was confirmed 30 months later. The case report published by Gonzalvez-Garcia et al. [149] in 2013 did not differ significantly from that of Cella et al. [144], but the application of autologous MSC together with β-tricalcium phosphate, demineralized bone matrix, and PRP yielded good results of cure after six months. A pilot study conducted by Voss et al. [150] in 2017 reported satisfactory healing after applying autologous MSC to the site where the necrotic bone was removed, and autologous thrombin was applied in 6 patients with MRONJ Stage 2, covering it with a collagen membrane. They mentioned that MSC transplantation with surgical treatment is a promising treatment method for the treatment of MRONJ. The most recent report by Santis et al. [151] showed a successful treatment result in two MRONJ patients by applying autologous MSC expanded ex vivo for 3–4 weeks together with a bone substitute. In contrast to the autologous MSC, studies in which adipose-derived MSC or allogenic bone marrow MSC is applied to the treatment of MRONJ are limited to animal studies.

In the treatment of MRONJ, the limitation of MSC therapy is that it is not known precisely whether the primary mechanism of MSC is due to bone regeneration associated with osteoblast differentiation (direct or indirect) or immunomodulatory properties, or both. Moreover, there are some drawbacks to MSC therapy, including a bothersome procedure of collecting MSC from other parts of the body and the need for additional equipment such as a centrifuge. However, this shortcoming already exists in the application of PRP or PRF, and it is expected to be sufficiently overcome. Lastly, similar to other treatment methods using graft, it is challenging to apply MSC as a sole treatment modality in the treatment of MRONJ. Instead, it seems to be an adjunctive treatment method that requires surgical treatment. However, despite these limitations, treatment using MSC is considered to be one of the most promising treatment methods along with teriparatide due to the regenerative potential of MSC itself.

## 5. Conclusions

MRONJ is currently a refractory disease that must be overcome in the maxillofacial area, and the gold standard treatment has still not been identified. Various treatment modalities have been developed, but they are insufficient to heal MRONJ lesions as a single treatment method. Teriparatide is in the spotlight as a new therapeutic agent and has shown potential as a single therapeutic solution for the MRONJ, but it needs to be verified through more clinical studies to establish itself as a standard treatment. Besides, many studies developing new cellular therapeutic approaches such as MSC therapy are ongoing. A new path of MRONJ treatment may be built through various combinations of existing treatment methods or those of the existing and the new treatment methods. In the management of the MRONJ, prevention is also crucial along with the treatment. For the prevention of MRONJ, patients scheduled to be administered antiresorptive or antiangiogenic agents should undergo thorough intraoral screening and appropriate dental care such as patient motivation and education to maintain oral hygiene, fluoride application, and chlorhexidine rinses. Moreover, multidisciplinary discussions with doctors in other medical fields and biological scientists are needed on topics such as the appropriate drug-free period of each triggered drug and the risk evaluation regarding the accumulation period. It is also believed that further research and studies for the development of effective treatment modalities for MRONJ are required, along with the cooperation between researchers and clinicians.

## Figures and Tables

**Figure 1 antioxidants-10-00680-f001:**
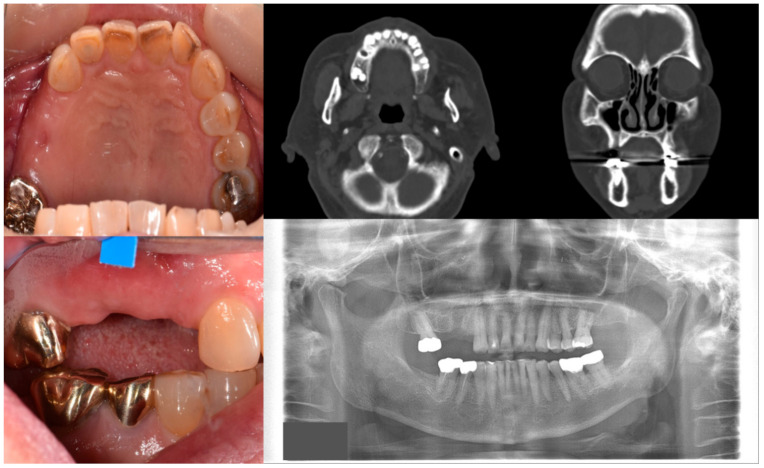
Case 1 with MRONJ Stage 0: Clinical photos, panoramic view, and computed tomography imaging showing non-significant findings on the maxillary right posterior area. An 81-year-old female patient with a history of taking oral alendronate for five years complained of pain after removing the right upper second premolar five months before. There were no specific findings other than slight soft tissue depression in the area and hard tissue defect presumed to be an extraction socket.

**Figure 2 antioxidants-10-00680-f002:**
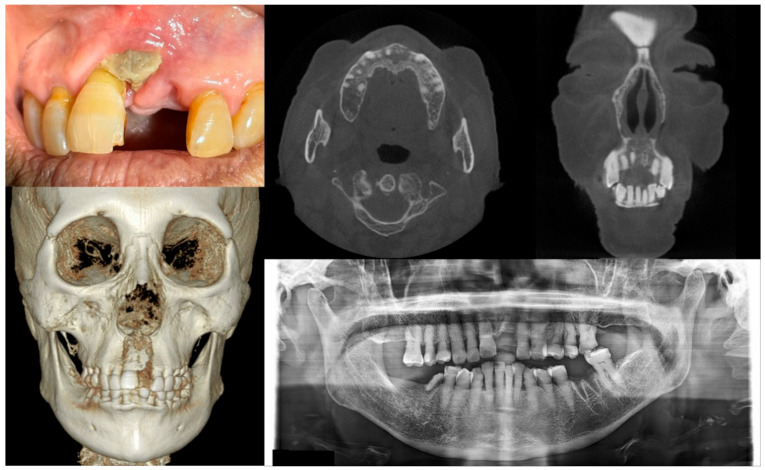
Case 2 with MRONJ Stage 1: Clinical photo, panoramic view, and computed tomography imaging including three-dimensional reconstruction showing exposed alveolar bone on the anterior maxillary area. An 89-year-old female patient with a history of oral alendronate for more than four years and intravenous administration of denosumab for one year showed the exposed bone after extracting the left upper central incisor three months ago. She reported no symptoms such as pain other than exposed bones. There was no sign of infection, including pus discharge on the exposed bone.

**Figure 3 antioxidants-10-00680-f003:**
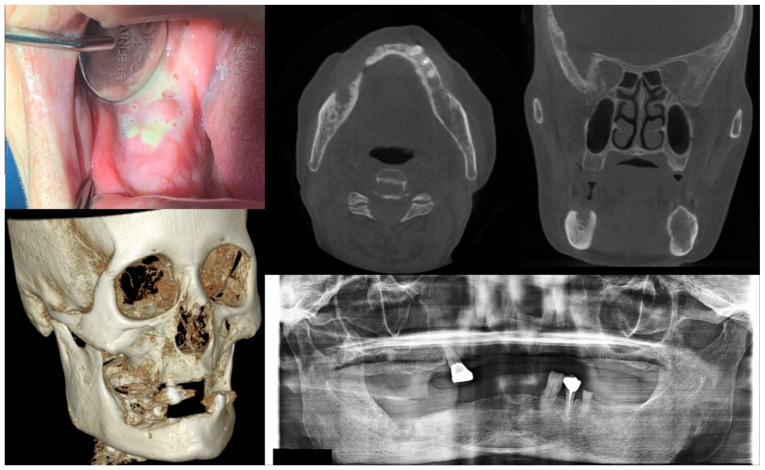
Case 3 with MRONJ Stage 2: Clinical photo, panoramic view, and computed tomography imaging including three-dimensional reconstruction showing exposed alveolar bone with pus discharge on the right mandibular posterior area. A 68-year-old female patient with a history of intravenous administration of ibandronate for eight years and intravenous administration of denosumab for the last year complained of facial swelling, severe pain, and pus discharge. She had undergone extraction of the right lower second molar at the private dental clinic to relieve the facial swelling and pain. Symptoms were relieved slightly after extraction but worsened again after two weeks. As a result of the examination, exposed bone and infection signs were observed.

**Figure 4 antioxidants-10-00680-f004:**
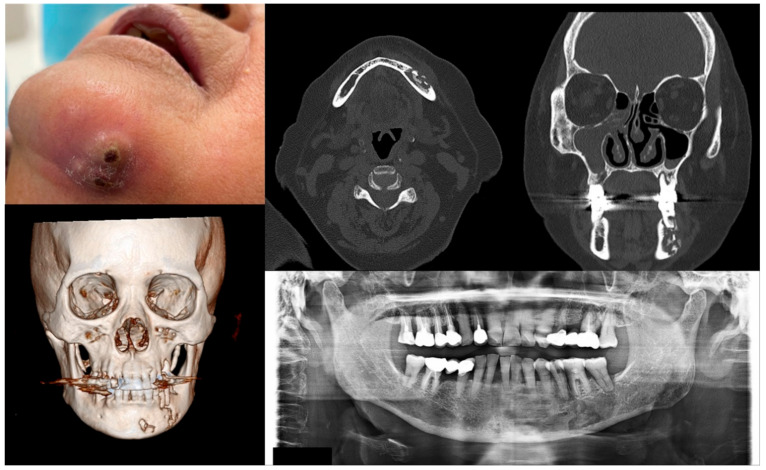
Case 4 with MRONJ Stage 3: Clinical photo, panoramic view, and computed tomography imaging including three-dimensional reconstruction showing extraoral fistula and necrotic bone on the mandibular left anteroposterior area. An 80-year-old female patient with a history of intravenous administration of ibandronate and denosumab for one year was presented with inflammation and skin fistula on the chin area. She had undergone periodontal treatment on the left lower posterior teeth three months ago due to a periodontal abscess, but she was referred for recurrence with facial swelling from a private dental clinic. Bony lesions involving the inferior border of the mandible and the presence of extraoral fistula were observed.

**Table 1 antioxidants-10-00680-t001:** Drugs are known to be associated with MRONJ.

Drugs	Type of Drug	Indications
Alendronate Clodronate Etidronate Ibandronate Pamidronate Risedronate Tiludronate Zoledronate	Bisphosphonates	Osteoporosis, Paget’s disease, bone metastases of malignancies, hypercalcemia of malignancy
Denosumab	Inhibitor of receptor activator of nuclear factor-kappa B ligand	Osteoporosis, bone metastases of malignancies, hypercalcemia of malignancy
Sunitinib Sorafenib	Tyrosine kinase inhibitors	Metastatic cancers (breast, renal, lung, colorectal)
Bevacizumab Rituximab	Vascular endothelial growth factor inhibitors	Glioblastoma, metastatic cancers (breast, renal, lung, colorectal)

**Table 2 antioxidants-10-00680-t002:** Staging of MRONJ [2].

Stage	Clinical Manifestation	Strategies for Management
At risk	No apparent necrotic bone	Patient education, no treatment required
0	No clinically necrotic or exposed bone with nonspecific clinical and radiographic findings	Systemic medication, including antibiotics and pain-killers when indicated
1	Exposed necrotic bone or intraoral fistula with no symptoms and no infection	Patient education, oral antibacterial rinse, close clinical follow-up
2	Exposed necrotic bone or intraoral fistula showing symptoms and signs associated with infection	Antibacterial oral rinse, systemic medication with antibiotics and pain-killers, debridement reducing the volume of necrotic bone for infection control
3	Exposed necrotic bone or intraoral fistula showing symptoms and signs associated with infection + exposed necrotic bone invading the inferior border of the mandible or the sinus floor of the maxilla and/or extraoral fistula and/or oroantral fistula	Antibacterial oral rinse, medication with antibiotics and pain-killers, surgical treatment including open debridement/resection for long-term control of symptoms and infection

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
