# Peer review of "Various Therapeutic Methods for the Treatment of Medication-Related Osteonecrosis of the Jaw (MRONJ) and Their Limitations: A Narrative Review on New Molecular and Cellular Therapeutic Approaches"

_antioxidants, 2021, doi:10.3390/antiox10050680_

Round 1

Reviewer 1 Report

In my opinion is an interesting, well-conducted and easy-to-read review of the literature. I congratulate the authors for this interesting work.

Reviewer 2 Report

This manuscript presents a review of treatment methods for medication-related osteonecrosis of the jaw. The manuscript is generally well-written, I have just a few methodological comments on the manuscript.

1. It is unclear if all relevant literature has been included in the review or what principles have guided the selection of included articles. I recommend clarifying whether the manuscript or a specific part of the manuscript, represents a systematic review with regard to the included literature. If this is the case, the manuscript or the relevant part of the manuscript should be made compliant with the PRISMA Statement and a completed PRISMA Statement checklist should be included with the manuscript. In any case, I recommend identifying the report in the title either as a systematic or narrative review. 

2. I also recommend specifying the procedure for the literature search (e.g. years considered, language, publication status, study design, databases, etc.).

Reviewer 3 Report

Thank you for giving be the opportunity to comment on this interesting narrative review.

The topic is interesting, the paper is well organized and written correctly.

I just have a few minor comments on it.

Please specify which drugs can cause ONJ and how ONJ might be prevented? 
